# Prevalence of glucose-6-phosphate dehydrogenase deficiency (G6PDd), CareStart qualitative rapid diagnostic test performance, and genetic variants in two malaria-endemic areas in Sudan

**Musab M. Ali Albsheer**[1,2], **Andrew A. Lover**[3], **Sara B. Eltom**[1], **Leena Omereltinai**[1], **Nouh Mohamed**[4], **Mohamed S. Muneer**[1,5], **Abdelrahim O. Mohamad**[5], **Muzamil Mahdi Abdel Hamid**[1]*

1 Department of Parasitology and Medical Entomology, Institute of Endemic Diseases, University of Khartoum, Khartoum, Sudan, 2 Faculty of Medical Laboratory Sciences, Sinnar University, Sennar, Sudan, 3 Department of Biostatistics and Epidemiology, School of Public Health and Health Sciences, University of Massachusetts-Amherst; Amherst, Massachusetts, United States of America, 4 Department of Parasitology and Medical Entomology, Faculty of Medical Laboratory Sciences, Nile University, Khartoum, Sudan, 5 Department of Biochemistry, Faculty of Medicine, University of Khartoum, Khartoum, Sudan

* mahdi@iend.org, mahdi@uok.edu

## Abstract

Glucose-6-phosphate dehydrogenase deficiency (G6PDd) is the most common enzymopathy globally, and deficient individuals may experience severe hemolysis following treatment with 8-aminoquinolines. With increasing evidence of *Plasmodium vivax* infections throughout sub-Saharan Africa, there is a pressing need for population-level data at on the prevalence of G6PDd. Such evidence-based data will guide the expansion of primaquine and potentially tafenoquine for radical cure of *P. vivax* infections. This study aimed to quantify G6PDd prevalence in two geographically distinct areas in Sudan, and evaluating the performance of a qualitative CareStart rapid diagnostic test as a point-of-care test. Blood samples were analyzed from 491 unrelated healthy persons in two malaria-endemic sites in eastern and central Sudan. A pre-structured questionnaire was used which included demographic data, risk factors and treatment history. G6PD levels were measured using spectrophotometry (SPINREACT) and first-generation qualitative CareStart rapid tests. G6PD variants (202 G>A; 376 A>G) were determined by PCR/RFLP, with a subset confirmed by Sanger sequencing. The prevalence of G6PDd by spectrophotometry was 5.5% (27/491; at 30% of adjusted male median, AMM); 27.3% (134/491; at 70% of AMM); and 13.1% (64/490) by qualitative CareStart rapid diagnostic test. The first-generation CareStart rapid diagnostic test had an overall sensitivity of 81.5% (95%CI: 61.9 to 93.7) and negative predictive value of 98.8% (97.3 to 99.6). All persons genotyped across both study sites were wild type for the G6PD G202 variant. For G6PD A376G all participants in New Halfa had wild type AA (100%), while in Khartoum the AA polymorphism was found in 90.7%; AG in 2.5%; and GG in 6.8%. Phenotypic G6PD B was detected in 100% of tested participants in New Halfa while in Khartoum, the phenotypes observed were B (96.2%), A (2.8%), and AB (1%). The

**Data Availability Statement:** The data underlying the findings are fully available without restriction at Open Science Framework (https://osf.io/k2nfw/).

**Funding:** Funding for this study was provided by the WHO/TDR/EMRO small grant scheme (SGS13/36; to MMAA), and by UMass (faculty research fund 2018-SPHHS; to AAL). The funders had no role in study design, data collection/analysis, decision to publish, or preparation of the manuscript.

**Competing interests:** The authors have declared that no competing interests exist.

African A- phenotype was not detected in this study population. Overall, G6PDd prevalence in Sudan is low-to-moderate but highly heterogeneous. Point-of-care testing with the qualitative CareStart rapid diagnostic test demonstrated moderate performance with moderate sensitivity and specificity but high negative predicative value. The two sites harbored primarily the African B phenotype. A country-wide survey is recommended to understand GP6PD deficiencies more comprehensively in Sudan.

## Author summary

Malaria is caused by five species of parasites; of these *Plasmodium falciparum* and *P. vivax* cause the majority of global morbidity and mortality. *Plasmodium vivax* infection is an emerging public health problem in sub-Saharan Africa, including Sudan. Primaquine and other 8-aminoquinolines including tafenoquine are the primary treatments to target the silent liver stage (hypnozoites) in *P. vivax* infections. However, these regimens can cause severe intravascular hemolysis in patients suffering from glucose-6-phosphate dehydrogenase deficiency (G6PDd). To support safe and efficacious use of primaquine, and potentially tafenoquine in Sudan, this study aimed to estimate the prevalence of G6PDd across two sites in Sudan using spectrophotometry and a qualitative CareStart rapid diagnostic test. Subsequent genetic analysis by PCR/RFLP and sequencing of G6PD genetic variants was performed. This survey found an overall prevalence was 5.5% (27/491; 30% of adjusted male median, AMM), and 27.3% (134/491; 70% of AMM) and 13.1% (64/490) by qualitative CareStart rapid diagnostic test. Important differences in distribution of genetic variants of G6PD were found across the two sites, and the African A- was not observed. In univariate analysis a few parameters showed significant association with G6PD deficiency. In conclusion the prevalence of G6PDd was low to moderate but heterogonous, and the first-generation qualitative CareStart rapid diagnostic test showed moderate performance in both males and females.

## Introduction

The Republic of Sudan has an extensive malaria burden with an estimated 1,600,000 cases (95% CI, 904,000 to 3,686,000) and approximately 5,000 deaths in 2018 [1]. Of this reported case total, approximately 1,300,00 (79.6%) were infections with *Plasmodium falciparum*; 140,000 (8.8%) were infections with *P. vivax*, and 190,000 (11.6%) were co-infections with *P. falciparum* and *P. vivax* [1]. A recent study from ten states across Sudan reported a prevalence of *P. vivax* in clinical malaria cases of 26.6% by PCR [2].

Sudan's national malaria control program is focused on malaria elimination, with insecticide-treated bed nets (ITN) as the primary preventive intervention, combined with indoor-residual spraying (IRS) in conjunction with early diagnosis and treatment [3]. In Sudan a 14-day primaquine regimen (0.25 mg/kg day) is currently used for a radical cure of *P. vivax* infection. To date, reports originating from Sudan on glucose-6-phosphate dehydrogenase deficiency (G6PDd) are limited [4–6]. The G6PD gene consists of 13 exons and 12 introns, which are located on the long arm of the X-chromosome at gene loci Xq28. More than 200 mutations have been reported [7], most of which are due to single nucleotide substitution [8].

The most common mutations across Sub-Sahara Africa are G6PD A− (202A/376G) and G6PD A (202G/376G or 202A/376A), both of which are characterized by a reduction in

enzyme activity [9]. In diverse global settings including Southern Europe, the Middle East, India, Papua New Guinea, and Iran theG6PD–Mediterranean variant (Med, C563T) is prevalent [5]. Globally, the most common wild type of the variant is G6PD*B (202G/376A), which exhibits normal enzyme activity and is present in the worldwide population [10]. While the G6PD enzyme has been well-studied in specific regions, large gaps remain, especially toward establishing an evidence-base for use of primaquine and other 8-aminoquinolines in control and elimination programs.

A major barrier to sustained control and elimination of *P. vivax* infections is the existence of hypnozoites (a quiescent parasite life stage) that remain dormant and undetectable in the host's liver tissue [11]. When these hypnozoites are reactivated by poorly-understood mechanisms, they can re-establish infections weeks or months later [12–14]. Current WHO guidelines for malaria programs are based on the use of primaquine to target hypnozoites; however, routine wide-scale use of a 14-day regimen for a radical cure necessitates G6PD testing whenever possible [15].

There are currently no laboratory or molecular methods to confirm carriage of hypnozoites. Therefore, treatment of symptomatic cases or mass drug administration is required to clear the hypnozoite reservoir. However, the relative risks of *P. vivax* infection in comparison to potential hemolytic harm from 8-aminoquinoline therapy are conditional on local endemicity and severity of local G6PD variants [16].

Primaquine and other 8-aminoquinolines have the potential to cause hemolysis that may range from mild and self-limiting, all the way to severe and requiring transfusion [17]. The WHO has developed a classification scheme for G6PD deficiencies which consists of four broad classes: class I (severe mutation); class II (intermediate mutation, 10% of normal G6PD function); class III (mild mutation, 10–60% of normal G6PD function); and class IV (asymptomatic, 60–100% of normal G6PD function). However, recent studies have suggested that these classes are broad and may be insufficient to accurately capture the diversity of G6PD enzymopathy, especially in females [15]. Although the need for G6PD testing before primaquine administration has been recognized, the health system in Sudan does not currently integrate G6PD testing into the malaria control program [18]. To address these challenges, and maximize the impact of the limited health budgets, there is a pressing need to measure and understand the prevalence of the well-described G6PDd alleles. To support these efforts, individuals who presented to clinics in two diverse malaria-endemic settings in Sudan were tested using biochemical and molecular methods. Additionally, the performance of first-generation qualitative CareStart G6PD rapid diagnostic tests (RDT) were evaluated for the screening of G6PD deficiency in point-of-care (POC) settings.

## Methods

### Ethics statement

This study received an ethical clearance from the Ethical Committee of Institute of Endemic Diseases, University of Khartoum (approval number: IRB/IEND 1/2014; date 10/6/2014). Written informed consent was signed by each study subject (adults) or guardians (children). UMass IRB determined the analysis of this study was exempt from human subject review (determination #1798; Jan. 2020).

### Study areas

This cross-sectional survey was conducted in two malaria-endemic areas in Sudan (Fig 1) from February to May of 2015. The first site was Gezira Slanj, rural Khartoum, in central Sudan (15°53′11.2″N 32°31′39.9″E), where malaria transmission is seasonal. Two annual peaks

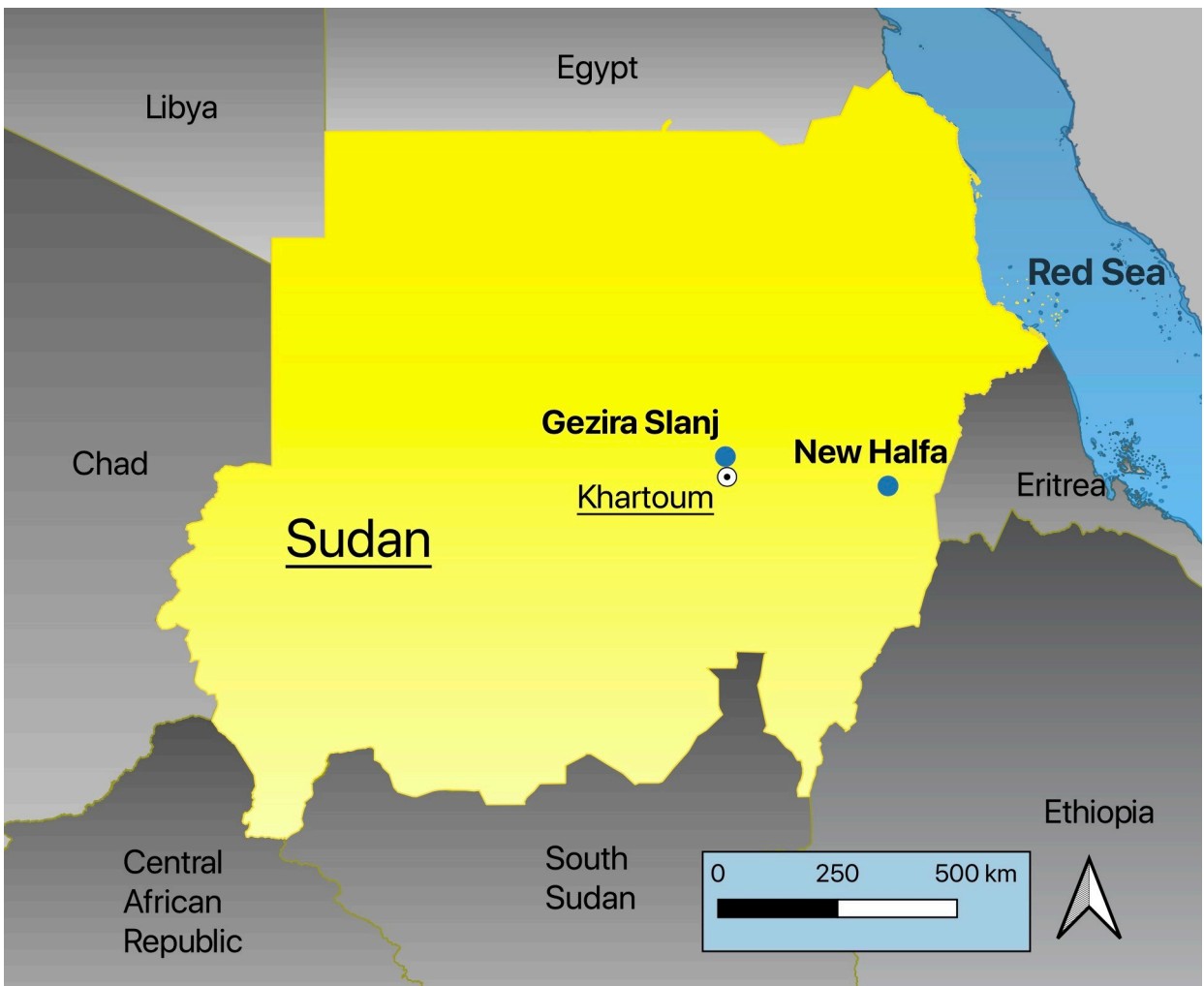

**Fig 1. Location of G6PD study sites, 2015, Sudan.** Map produced in QGIS [QGIS Development Team, 2021. QGIS Geographic Information System. Open-Source Geospatial Foundation. http://qgis.org]; basemaps [https://www.diva-gis.org/Data] and [Flanders Marine Institute (2021): MarineRegions.org. Available online at www.marineregions.org. Consulted on 2021-08-21].

of transmission occur in this area: one following the rainy season (July-October) while the other occurs during the temperate months (December-February). The second site was New Halfa town, in Eastern Sudan (15°20′N 35°35′00″E). There is mesoendemic malaria transmission in this region, with a single peak during the rainy season (July-October). Both study sites are co-endemic for *P. falciparum* and *P. vivax* [2,4].

## Study population, study power and sampling strategy

The study population consisted of apparently healthy subjects who were accompanying patients attending primary health units within the study areas. Women who were pregnant or lactating, and all persons seeking care were excluded. Convenience sampling was used to invite participants to give informed consent to participate in the study. The study was powered for an assumed G6PD prevalence of 10%; a sample size of 450 inclusions provides a 95% CI of 7.4 to 13.1% (binomial exact), providing sufficient precision to inform policy and future research.

A total of 557 subjects were recruited across both study sites. Essential demographic characteristics were collected using a structured questionnaire, including basic demographic data, linguistic group, education, occupation, medical history, blood transfusion history, and detailed drug history, including adverse drug reactions. Venous blood samples were collected in EDTA and heparin tubes using aseptic conditions. The EDTA samples were subsequently used for hemoglobin estimation and DNA extraction, while the heparin-preserved samples were used for G6PD activity measurement within 30 minutes of blood collection.

## G6PD activity measurement and qualitative CareStart point of care (POC) testing

The gold standard biochemical G6PD measurements was collected according to published methods [19]; using a SPINREACT kit with spectrophotometric reading at 340nm and 25˚C (BioSystems BTS 310) to measure G6PD activity in heparinized venous blood, as per manufacturer instructions. Normal and deficient G6PD controls provided with the kit were run daily prior to all clinical sample measurements (Ref: 100 252 0, SPINREACT, Spain).

The CareStart G6PD point-of-care tests (Access Bio, Catalog # G0221; 2014 version) were performed according to the manufacturer instructions [20]. This is a qualitative test, providing either normal (purple) or a deficient (colorless) result. The spectrophotometry and CareStart tests were performed by two independent readers who were blinded to all other results. All sample analyses were performed within 30 minutes of blood collection. Extreme readings were recorded but not re-analyzed, and addressed in the statistical analysis.

## G6PD population-level activity

In this study the adjusted male median (AMM) activity of enzyme activity was used as the reference cutoff value for the determination of deficiency status within the study population [19]. Briefly, the population reference value was defined as the population median after removal of all males with measured activity <10% of the median; and an activity cutoff of <30% of this adjusted male median was used to define G6PDd in this population as a whole [19]. A more stringent cutoff at <70% of the AMM was also evaluated to support potential tafenoquine implementation.

## DNA extraction and G6PD variants genotyping

DNA extraction was performed using the standard phenol-chloroform method [21], and genotyping was conducted according to previously published methods [22] with some minor modification in annealing temperature of PCR. PCR was performed for exons 4 and 5 of the G6PD gene using previously published primers [22]. Briefly, PCR reactions were performed in total volume of 20 μl containing genomic DNA, 10pmol of the primers and PCRPreMix (i-Taq Intron Biotechnology, South Korea) with a thermocycler(SensQuick Inc., Germany). The adjusted annealing temperatures were 61.3˚ C and 60.3˚ C for the G6PD exon 4 and exon 5, respectively.

PCR products (5 μl) were digested with 1 U of *Nla*III and *Fok*I endonucleases (New England BioLabs Inc., UK) for (G6PD G202A) and (G6PD G376A) and incubated at 37˚C overnight, analyzed on 2% agarose gel and visualized using a gel documentation system (BDA compact, Biometra, Germany). Restriction fragment length polymorphism (RFLP) results were confirmed by commercial DNA sequencing of exons 4 and 5 (Macrogen Inc, Netherlands) on a subset 60 samples, chosen to represent diverse PCR/RFLP patterns.

## Statistical analysis

The population-level values for G6PD activity were compared using k-sample nonparametric median tests, and summarized as medians due to non-normal distributions, with bootstrapped 95% CIs. Categorical variables were compared using $\chi^2$ tests. The adjusted male median breakpoint for G6PDd was determined using the method of < 30% (or <70% where indicated) enzyme activity of the adjusted male median [23]. Logistic models were used to assess factors associated with G6PDd, from surveyed questionnaire items. These factors were examined using univariable models; all factors significant at p < 0.20 were taken forward to multivariable models; Akaike and Bayesian Information Criteria (AIC/BIC) were used to assess model parsimony. Logistic regression model fit was assessed using the le Cessie-van Houwelingen-Copas-Hosmer unweighted sum of squares test and Tjur's R2 [24,25]. Analyses were performed with Stata 14 or 16 (College Station, TX; USA) and R software [26]; all tests were two-sided, with $\alpha$ = 0.05. While these data had generally low levels of missing values, multiple imputation (MI) was used to ensure robust results, and to minimize potential biases from complete-case analysis in logistic regression. All risk factors significant at p < 0.20 were imputed using MI via chained equations, using data augmentation for variables that exhibited complete separation with a total of 50 imputations. Adequacy of the imputed datasets was assessed using standard recommendations [26]. Age was missing for a single observation, which led to instability in multiple imputations, and this observation was removed for final models.

## Results

A total of 557 persons consented and were enrolled across both study sites. A subset of these participants had missing timepoints for spectrophotometry, or biologically implausible values in biochemical testing. These 66 individuals (32 females, and 34 males), did not differ from the overall study population by sex, (p = 0.67; $\chi^2$ test), but were all hospital cases at one of the sites (both $\chi^2$ tests, p < 0.001). Due to missing outcome data, these were removed from further analysis, providing a final study population of 491 persons. The overall population attributes and selected survey are presented in Table 1. Detailed population characteristics and other survey responses can be found in the Supplemental information (S1 Table).

### Biochemical testing

The measured G6PD activities for both sites by spectrophotometry are presented in Table 2. Recorded values ranged from 0.22 to 13.0 U/g Hb, with a median of 4.27 (95% CI: 3.95 to 4.59) U/g Hb. There was a statistically significant difference in median enzyme activity by study site (p < 0.001); but not by sex (p = 0.89), using a nonparametric k-means test (Table 2). The distributions of measured activities are shown by study site (Fig 2) and by sex (Fig 3). From these measurements, the corresponding population adjusted male median value was 4.23 U/g Hb.

Of the 491 subjects, 27 subjects (5.5%; 95% CI: 3.7 to 7.9) had severe or moderately severe G6PDd, with a measured enzyme activity <30% of this adjusted male median (1.27 U/g Hb). Twenty-four of these were from the New Halfa site, including 16 females, while three were from Khartoum, including two females (Table 3). One individual (a female in Khartoum) had extremely low activity (< 10% of the AMM). Using the more stringent threshold of70% of AMM, a total of 134 (27.3%) persons would be classified as G6PDd. (Table 3).

### Point-of-care (POC) qualitative rapid test (CareStart) and concordance with biochemical testing

A total of 491 persons were tested for G6PDd using CareStart POC tests. Of these tests, 490 results were available for analysis; one test gave an invalid reading. A total of 64 subjects

**Table 1. Study population overview G6PD surveys, New Halfa and Khartoum Sudan, 2015 (N = 491).**

| Characteristic | Value | n (% of total) |
|---|---|---|
| Study site | New Halfa | 327 (66.6%) |
| | Khartoum | 164 (33.4%) |
| | Missing | 0 (-) |
| Data source | Private clinic | 125 (25.5%) |
| | Public hospital | 360 (73.3%) |
| | Missing | 6 (1.2%) |
| Sex | Female | 252 (51.3%) |
| | Male | 239 (48.7%) |
| | Missing | 0 (-) |
| Age, mean (SD) | | 34.2 (± 15.6) |
| | Missing | 0 (-) |
| Anemia (any; WHO standards) | No | 264 (53.8%) |
| | Yes | 227 (46.2%) |
| | Missing | 0 (-) |
| Weight in kg, mean (SD) | | 59.7 (± 18.4) |
| | Missing | 7 (1.4%) |
| Bilirubin in mg/dL, median (IQR) | | 0.3 (0.1, 0.6) |
| | Missing | 0 (-) |
| Hemoglobin (Hb), mean (SD) | | 12.5 (± 2.4) |
| | Missing | 0 (-) |
| G6PD activity (U/g Hb) by spectrophotometry, median (95% CI) | | 4.3 (4.0 to 4.6) |

(13.1%, 95% CI: 10.2 to 16.4%) were G6PDd according to the CareStart test. There were significant differences by geographical location (p = 0.031) but not by sex (p = 0.258) via $\chi^2$ tests. Using cutoffs of 30% and 70% of the AMM (1.27and 2.96 U/g Hb respectively), the performance of RDT compared to the gold-standard test is shown in Table 4 and Fig 4. The overall sensitivity was moderate at 81.5% (95% CI: 61.9 to 93.7), with comparable values in males and females (78% vs 83% with wide confidence intervals). No differences were observed in specificity and negative predicative value between male (91% and 99%) and female populations (91% and 99%).

## Factors associated with G6PDd

In bivariate analysis, multiple factors showed a significant association with G6PDd status (less than 30% AMM), including study site (odds ratio of G6PDd Khartoum relative to New Halfa = 0.23 (95% CI: 0.07 to 0.79), p = 0.02); data source (public hospital relative to private clinic, OR = 0.30 (95% CI 0.14 to 0.65), p = 0.002), and any self-reported recent antibiotic use (OR = 2.43 (95% CI: 1.1 to 5.5), p = 0.031). However, after adjustment for covariates in multivariable analysis, the only factor that remained statistically significant was the subject's weight

**Table 2. Summary of G6PD activities (with 95% CIs) by biochemical testing, total study population, and in New Halfa and Khartoum, Sudan.**

| | G6PD activity, U/g Hb (median, 95% CI) | | |
|---|---|---|---|
| | Overall study population (N = 491) | New Halfa (n = 327) | Khartoum (n = 164) |
| **Total tested population** | 4.3 (4.0 to 4.6) | 3.69 (3.3 to 4.0) | 5.0 (4.6 to 5.4) |
| **Males only** | 4.2 (3.8 to 4.6) | 3.8 (3.3 to 4.2) | 5.0 (4.6 to 5.5) |
| **Females only** | 4.3 (3.9 to 4.7) | 3.7 (3.2 to 4.1) | 4.8 (4.4 to 5.3) |

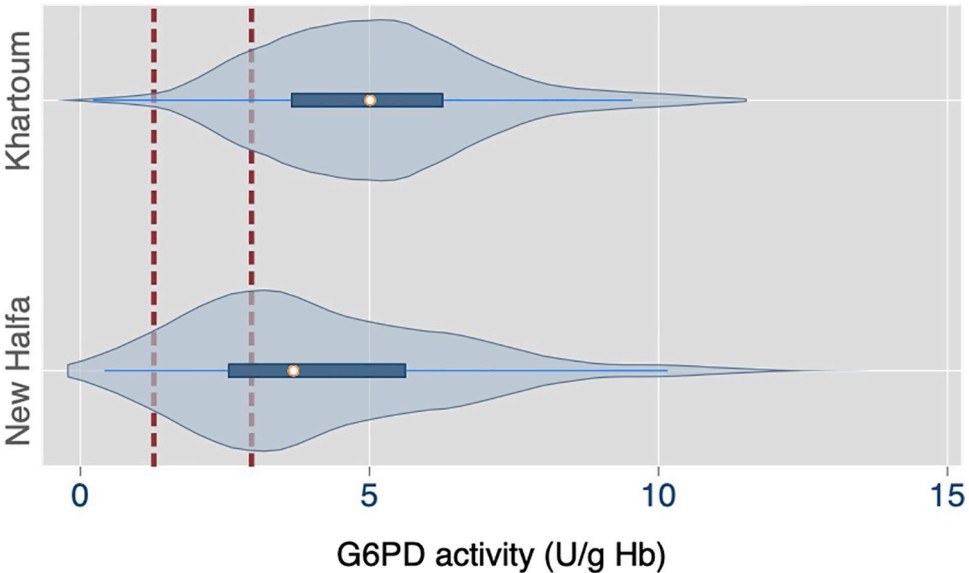

N = 491; Reference lines: 1.27 (30% AMM) & 2.96 (70% AMM).

**Fig 2. Distribution of G6PD activities in study population, by gender (n = 491).**

in kilograms, with an adjusted OR of 0.97 (95% CI 0.95 to 0.99; p = 0.01). This indicates that each kg increased body weight lowers the odds of being G6PD deficient (<30% AMM), by approximately 3% (See S1 Table).

## G6PD genotyping

PCR/RFLP analysis was performed for all 491 enrolled subjects; 397 of which were successfully amplified for the G6PD gene and met data quality thresholds. Considering the G202A G6PD

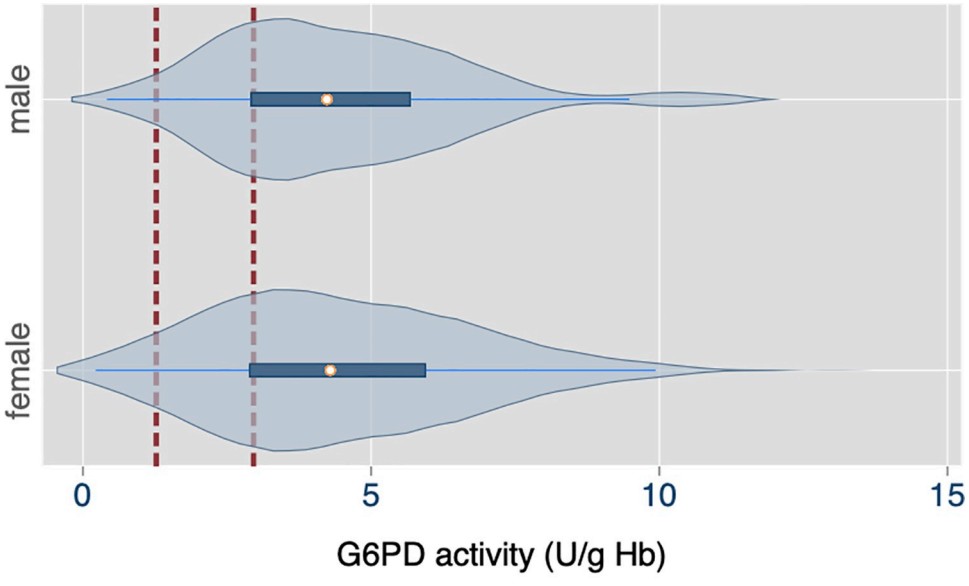

N = 491; Reference lines: 1.27 (30% AMM) & 2.96 (70% AMM).

**Fig 3. Distribution of G6PD activities in study population, by study site (n = 491).**

**Table 3. Prevalence of G6PD deficiency in surveyed populations by biochemical testing (SPINEACT test) (adjusted male median = 4.23 U/g Hb), New Halfa and Khartoum Sudan, 2015.**

| Study site | Sex | Total | G6PD deficient 30% of adjusted male median | G6PD deficient 70% of adjusted male median |
|---|---|---|---|---|
| Khartoum | F | 91 | 2.2% (95% CI: 0.01 to 7.7) (n = 2) | 13.1% (7.0 to 21.9) (n = 12) |
|  | M | 73 | 1.4% (0.01 to 7.4) (n = 1) | 12.3% (5.8 to 22.1) (n = 9) |
| New Halfa | F | 161 | 9.9% (5.8 to 15.6) (n = 16) | 35.4% (28.0 to 43.3) (n = 57) |
|  | M | 166 | 4.8% (2.1 to 9.3) (n = 8) | 33.7% (26.6 to 41.5) (n = 56) |
| Total |  | 491 | 5.5% (2.7 to 63) (n = 27) | 27.3% (23.4 to 31.4) (n = 134) |

gene polymorphisms across both study sites, all persons tested were GG homozygotes (100% GG genotype frequency). For G6PD A376G, the following genotypes were detected: AA (n = 235; 100%) in New Halfa; AA (n = 147; 90.7%), GG (n = 11; 6.8%), AG (n = 4; 2.5%) in Khartoum (Table 5).

Among the females examined in New Halfa (n = 122), the G6PD A376G genotype was entirely AA (100%), whereas the distribution in Khartoum (n = 90) was AA (n = 81; 90%); GG (n = 5; 5.6%); AG (n = 4; 4.4%) (Table 5); there was a statistically significant difference in the distribution of A376G polymorphisms by geographic location (p < 0.001; $\chi^2$ test). Of these participants with non-wild-types, the associated G6PD enzyme activities are shown in Fig 5.

In analysis of the 27 G6PD deficient subjects, 26 carried B variants, and one homozygous female carried an A variant. In this population, the A- variant was not detected; however, the Mediterranean variant was not tested for in this study. The genotype of all 27 G6PD deficient subjects was AA for the G202A polymorphism. For the A376G polymorphism, 26 subjects had the AA genotype while one subject carried the GG genotype.

**Table 4. Concordance between biochemical testing and qualitative CareStart results, New Halfa and Khartoum Sudan.**

|  | Biochemical testing (30% of adjusted male median) | | |
|---|---|---|---|
| CareStart qualitative result | Deficient | Not deficient | total |
| Deficient | 22 | 42 | 64 |
| Not deficient | 5 | 421 | 426 |
| total | 27 | 463 | 490 |

**Overall (N = 490) Percent (95% CI)**
 Sensitivity 81.5% (61.9–93.7); Specificity 90.9% (87.9–93.4); PPV 34.4% (22.9–47.3); NPV 98.8% (97.3–99.6).

**Males only (n = 239)**
Sensitivity 77.8% (40.0–97.2); Specificity 91.3% (86.9–94.6); PPV 25.9% (11.1–46.3); NPV 99.1% (96.6–100).

**Females only (n = 251)**
Sensitivity 83.3% (58.6–96.4); Specificity 90.6% (86.1–94.0); PPV 40.5% (24.8–57.9); NPV 98.6% (96.0–99.7).

|  | Biochemical testing (70% of adjusted male median) | | |
|---|---|---|---|
| CareStart qualitative result | Deficient | Not deficient | total |
| Deficient | 31 | 33 | 64 |
| Not deficient | 103 | 323 | 426 |
| total | 134 | 356 | 490 |

**Overall (N = 490)**
Sensitivity 23.1%; (16.3–31.2); Specificity 90.7% (87.2–93.5); PPV 48.4% (35.8–61.3); NPV 75.8% (71.5–79.8).

**Males only (n = 239)**
Sensitivity 16.9%; (8.8–28.3); Specificity 90.8% (85.5–94.7); PPV 40.7% (22.4–61.2); NPV 74.5% (68.1–80.2).

**Females only (n = 251)**
Sensitivity 29.0%; (18.7–41.2); Specificity 90.7% (85.5–94.5); PPV 54.1% (36.9–70.5); NPV 77.1% (70.9–82.6).

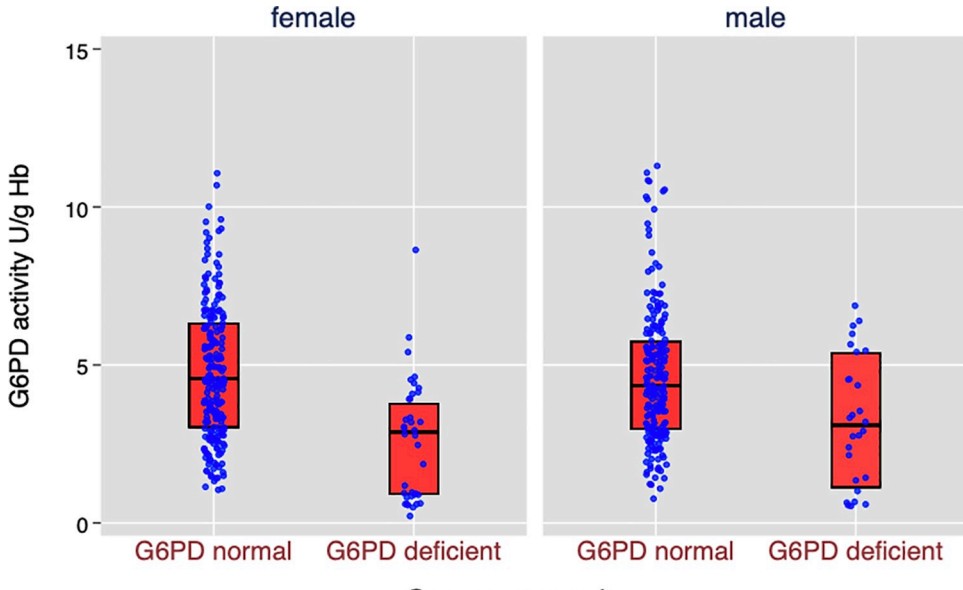

**Fig 4. Distribution of G6PD activities, by CareStart results (n = 490).**

## Discussion

Sudan's national malaria control program is focused on malaria elimination with insecticide-treated bed nets (ITN) as the primary preventive intervention. ITN is combined with indoor residual spraying and early diagnosis and treatment, mainly with artemisinin-based combination therapies (ACTs). Reducing the burden of malaria from *P. falciparum* (single low-dose), and *P. vivax* may require expansion of primaquine use. However, there is very limited evidence-based data on the safety of such health programs for patients with G6PDd. While extensive studies have been done throughout Southeast Asia to examine the prevalence and diversity of G6PD deficiencies [27], studies reporting data from Africa are emerging and limited. Studies to date have assessed G6PD in Ghana [28], Mauritania [29], Eritrea [30], and Ethiopia [31,32]. However, there have been no detailed studies from Sudan.

This study reported the frequencies of G6PDd in two sites, their genotypic and phenotypic characterization, and the results of evaluation of a qualitative POC test. The overall G6PDd

**Table 5. Comparison of genotypic data by study site, Sudan (n = 397).** (note: WT = wild type).

| Sex | Genotype | Phenotype Label | Interpretation | G6PD Activity, U/g Hb Median (95% CI) | New Halfa n = 235(%) | Khartoum n = 162 (%) |
|---|---|---|---|---|---|---|
| **M** | 202G/376A | B | Hemizygous WT | 4.1 (3.5–4.6) | 113 (48.1) | 66 (40.7) |
| | 202G/376G | A | Hemizygous A | 4.8 (3.7–5.9) | 0 (0.0) | 6 (3.7) |
| | 202A/376A | | | - | 0 (0.0) | 0 (0.0) |
| | 202A/376G | A- | Hemizygous A- | - | 0 (0.0) | 0 (0.0) |
| **F** | 202G, 202G/376A, 376A | B/B | Homozygous WT | 4.3 (3.9–4.7) | 122 (51.9) | 81 (50.0) |
| | 202A, 202G/376A, 376A<br>202G, 202G/376G, 376A | A/B | Heterozygous A | 6.8 (4.3–9.4) | 0 (0.0) | 4 (2.5) |
| | 202G, 202G/376G, 376G<br>202A, 202A/376A, 376A | A/A | Homozygous A | 4.7 (1.7–7.6) | 0 (0.0) | 5 (3.1) |
| | 202A, 202G/376G, 376A | A-/B | Heterozygous A- | - | 0 (0.0) | 0 (0.0) |
| | 202A, 202A/ 376G, 376G | A-/A- | Homozygous A- | - | 0 (0.0) | 0 (0.0) |

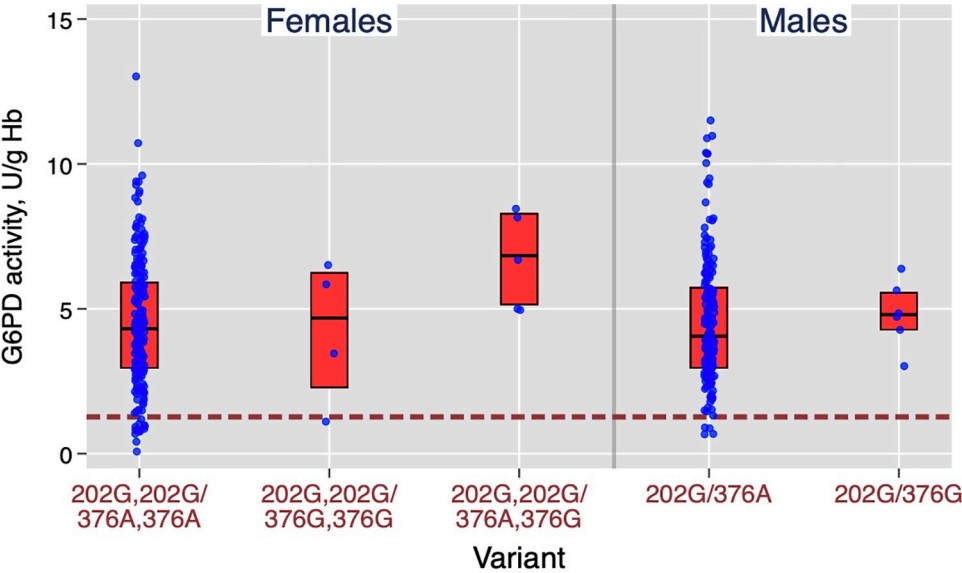

Cutoff = 1.27 U/g Hb (30% AMM)

**Fig 5. Distribution of G6PD activities, by genotyping results (n = 397).**

prevalence was estimated at 5.5% by the spectrophotometry with significantly higher prevalences New Halfa. This likely related to a differing ethnic composition of the population in New Halfa [33]. This reported G6PDd prevalence is considered "moderate" according to WHO standards [34]. Testing with the CareStart POC test provided a higher estimated prevalence, which is consistent with other studies [35–38]. Earlier reports of G6PD in Sudan focused on enzyme isoforms only, but did not provide associated frequencies [6,39]; and a recent study conducted in Northern Sudan reported on polymorphic genetic variants of the enzyme [5].

The reported prevalence in the current study is broadly consistent with results from Ethiopia, Egypt and Eritrea [31,40,30]. However, this G6PDd prevalence is lower than those reported from West African settings, including Senegal (12%) [41], Sierra Leone (11.3%) [42] and Burkina Faso (9.3%) [43]. The qualitative CareStart test performed moderately well in this study compared with the reference method. This is in concordance with previous reports from Cambodia [35], Philippines [36], Ghana [37], and Indonesia [38]. These results highlight some limitations in the qualitative CareStart test for the detection of persons with low or moderate enzyme levels, with a sensitivity of 82%. We did not observe any major differences between performance in males and females, as has been observed in other studies, e.g. Cambodia [35]. The limitations of non-quantitative test for screening of females occurs heterozygous females produce heterogeneous levels of the enzyme ("Lyonization"). This heterogeneity is due to random activation of one X-chromosome during embryogenesis, resulting in varying proportions of deficient and normal red blood cells in different heterozygotes females, which ultimately leads to a range of observed G6PD activities ranging from very low to near normal levels [20]. These findings agree with the field evaluation of G6PD CareStart RDT in Cambodians [35]. In the present study, the moderate sensitivity of CareStart POC as a qualitative test may not be adequate for diagnosis for G6PDd in these populations. The present study found that the only factor associated with G6PD activity in adjusted models was bodyweight in kg, with an adjusted OR of 0.97 (95% CI: 0.95 to 0.99), p = 0.01). While similar associations have been found in murine models [44], this finding is likely of limited public health utility.

The most common G6PD phenotypic variant in this study was G6PD B (96.2%, 382/397) which is consistent with results from a previous study in northern Sudan [5]. Similarly, other reports from Eritrea and Burkina Faso showed a high frequency of the G6PD B variant (87.5% and 74.5%, respectively) [30,43]. Lower frequencies of the B variant were reported in surveys from Sierra Leone, Ghana, and Cameroon [45]. The African variant (A) was detected in (2.8%), 11/397) of the study subjects, which is consistent with a previous study from northern Sudan which reported that among 207 participants, 90.3% carried the B variant, 1.4% carried the A; and 8.3% were not assigned a phenotype.[5]. The current study found the G6PD AB variant was present in a limited number of study subjects (1.0%; 4/397), all of which were female. Finally, we found no evidence for the G6PD A- isoform, in agreement with other studies in Sudan and recent reports from Eritrea [30], and Ethiopia [32]. Similar to the prevalence of G6PDd, the genetic variants also showed significant variation between the two study sites which may be related to different ethnic composition of these populations.

The 27 deficient subjects also suggest that G6PD B is the most prevalent variant in this sampled population, and the single A variant was a homozygous female. However, this study cannot exclude the presence of the Mediterranean and other variants which may co-exist with the ones reported here, and further surveys are required to capture the full diversity of G6PD variants in Sudanese populations.

As expected, there were large differences in measured enzyme activity by the genotype of males (hemizygous) and females (homozygous and heterozygous), as were differences in the distribution of the enzyme activity in the females, with heterozygous females having values ranging from low to normal enzyme activity [20]. However, there are several unusual features of the measured enzyme activities in this study. Specifically, the male population does not show the expected bimodal distribution as observed in most other settings [46,47], and surprisingly, a greater number of G6PDd women than G6PDd men were detected in biochemical testing. Few published results have appeared with the spectrophotometric test kit used in this study, which also required temperature adjustment as per manufacturer instruction. Together, these unusual findings suggest a critical role for follow up studies to understand this complex array of biochemical results.

In conclusion, this study provides the first large-scale data on the prevalence of G6PDd, and associated enzymatic activity and predictors in individuals by using phenotypic and molecular tests in endemic areas of *P. vivax* malaria in Central and Eastern parts of Sudan. The most common G6PD phenotypic variant in this study was B including the deficient subjects with absence of A-. The limited number of severely deficient G6PD activities across these two study sites (one female), suggests hemolysis risks may be somewhat limited at a population-level where milder variants predominate [48]. The lack of a clear association between genotype and measure enzyme activity warrants further verification with more diverse samples from other sites, and with more detailed genotyping. Other additional genetic variants that were not analyzed in this study may also have important associations with G6PD deficiencies in Sudan.

It should be emphasized that Sudan was highlighted as an area with unusually large uncertainty in earlier statistical models, and these estimates were based on the landmass before the independence of South Sudan. The impact of this areal unit problem is difficult to assess, and this current study greatly expands the evidence-based for G6PD-related policy in Sudan.

The use of primaquine within the public health sector in Sudan is a national policy. However, access is limited in some areas, which is a common scenario in many vivax-endemic countries [49].

Results from this study highlight the heterogeneity of G6PD level in the Sudanese population, and the good performance of the CareStart test for detection of moderate-to-severe

G6PD deficiency. Important gaps remain to ensure safe and routine radical cure of *P. vivax* infections towards the Sudanese Ministry of Health's malaria elimination goals.

In conclusion, based upon data from these two sites, the prevalence of G6PDd using the 30% AMM is somewhat lower than previous estimates. However, larger population-based studies and operational research on the use of point-of-care G6PD testing are needed to ensure the safe use of primaquine and potentially tafenoquine in diverse Sudanese populations.

## Supporting information

**S1 Table. Detailed population characteristics, New Halfa and Khartoum, Sudan 2015 (N = 491).**
(DOCX)

**S2 Table. Multivariable risk factors for Glucose-6-phosphate-dehydrogenase deficiency, Sudan [Deficiency defined as activity less than 30% of adjusted male median (AMM)].**
(DOCX)

## Acknowledgments

We would like to thank the volunteers who participated in this study. Our thanks are extended to all field staff and to Ms. Hana Abdallah Osman for her excellent technical contribution at the study initiation phase. The authors are also grateful to Dr. Ghasem Zamani and Dr. Ahmed Mandil from WHO/EMRO/TDR, Cairo, Egypt for their support and advice. We would like to thank Dr. Young S. Hong from Access Bio Inc., USA for providing the CareStart rapid tests.

## Author Contributions

**Conceptualization:** Musab M. Ali Albsheer, Muzamil Mahdi Abdel Hamid.

**Data curation:** Andrew A. Lover.

**Formal analysis:** Andrew A. Lover.

**Funding acquisition:** Musab M. Ali Albsheer, Muzamil Mahdi Abdel Hamid.

**Investigation:** Musab M. Ali Albsheer, Muzamil Mahdi Abdel Hamid.

**Methodology:** Musab M. Ali Albsheer, Sara B. Eltom, Nouh Mohamed.

**Project administration:** Muzamil Mahdi Abdel Hamid.

**Resources:** Musab M. Ali Albsheer, Muzamil Mahdi Abdel Hamid.

**Software:** Andrew A. Lover, Leena Omereltinai, Mohamed S. Muneer.

**Supervision:** Musab M. Ali Albsheer, Muzamil Mahdi Abdel Hamid.

**Validation:** Andrew A. Lover, Abdelrahim O. Mohamad.

**Visualization:** Andrew A. Lover.

**Writing – original draft:** Musab M. Ali Albsheer, Andrew A. Lover, Abdelrahim O. Mohamad, Muzamil Mahdi Abdel Hamid.

**Writing – review & editing:** Musab M. Ali Albsheer, Andrew A. Lover, Abdelrahim O. Mohamad, Muzamil Mahdi Abdel Hamid.

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
