## [Decision Letter · Decision Letter 0]

27 Jan 2021

Dear Dr Abdel Hamid,

Thank you very much for submitting your manuscript "Prevalence of Glucose-6-phosphatedehydrogenase deficiency (G6PDd), CareStart™ rapid diagnostic test performance and genetic variants in two malaria-endemic areas in Sudan" for consideration at PLOS Neglected Tropical Diseases. As with all papers reviewed by the journal, your manuscript was reviewed by members of the editorial board and by several independent reviewers. In light of the reviews (below this email), we would like to invite the resubmission of a significantly-revised version that takes into account the reviewers' comments. 

We cannot make any decision about publication until we have seen the revised manuscript and your response to the reviewers' comments. Your revised manuscript is also likely to be sent to reviewers for further evaluation.

Sincerely,

Wuelton Marcelo Monteiro, Ph.D.

Associate Editor

Marcelo Ferreira

Deputy Editor

Reviewer's Responses to Questions

**Key Review Criteria Required for Acceptance?**

**Methods**

-Are the objectives of the study clearly articulated with a clear testable hypothesis stated?

-Is the study design appropriate to address the stated objectives?

-Is the population clearly described and appropriate for the hypothesis being tested?

-Is the sample size sufficient to ensure adequate power to address the hypothesis being tested?

-Were correct statistical analysis used to support conclusions?

-Are there concerns about ethical or regulatory requirements being met?

Reviewer #1: The spectrophotometry procedures require further details:

• Was venous or capillary blood used?

• How were samples stored prior to spectrophotometry?

• What spectrophotometry assay was used and at what temperature?

• Specify the type of spectrophotometer used

• How was Hb measured?

Are the objectives of the study clearly articulated with a clear testable hypothesis stated?

• While the objectives are clearly formulated, the article does not present a hypothesis (not needed due to observational character).

Is the study design appropriate to address the stated objectives?

• Yes

Is the population clearly described and appropriate for the hypothesis being tested?

• Yes

Is the sample size sufficient to ensure adequate power to address the hypothesis being tested?

• Kindly add a sample size calculation.

Were correct statistical analysis used to support conclusions?

• I assume that what the authors call the trimmed male median is better known as the adjusted male median (if correct, kindly stick to the latter term which is much more broadly used). It would be excellent if the authors in addition to a 30% threshold could add a 70% threshold to their considerations. The latter is highly relevant for the future introduction of tafenoquine and to identify heterozygous females.

Can the authors provide more details in the statistics section? Rather than stating that continuous data were compared “using non-parametric median tests”, name them. The authors next state that categorical variables were compared using the Fishers exact test, however for larger samples a Chi2 test is more appropriate, for paired samples as would be expected from categorized results from spec vs. the RDT the McNemars test for correlated proportions is more suitable.

Are there concerns about ethical or regulatory requirements being met?

• No

Reviewer #2: The final number of G6PD deficient cases do not support an association to weight claimed by the authors, but this is called out in my feedback to the authors

Reviewer #3: (No Response)

**Results**

-Does the analysis presented match the analysis plan?

-Are the results clearly and completely presented?

-Are the figures (Tables, Images) of sufficient quality for clarity?

Reviewer #1: Does the analysis presented match the analysis plan?

• Yes

Are the results clearly and completely presented?

• Please provide details on the delay between sample collection and measurement by spectrophotometry

o Does delay correlate to G6PD activity?

Are the figures (Tables, Images) of sufficient quality for clarity?

• Yes

Reviewer #2: The diagnostic performance analysis is not complete and the contingency tables are either incorrect, the sensitivity is wrong

Tables can be made clearer.

Reviewer #3: (No Response)

**Conclusions**

-Are the conclusions supported by the data presented?

-Are the limitations of analysis clearly described?

-Do the authors discuss how these data can be helpful to advance our understanding of the topic under study?

-Is public health relevance addressed?

Reviewer #1: Are the conclusions supported by the data presented?

• No kindly see “general comments” below

Are the limitations of analysis clearly described?

• No, kindly add a limitations section

Do the authors discuss how these data can be helpful to advance our understanding of the topic under study?

• Yes

Is public health relevance addressed?

• Yes, however the authors genotyped mostly for variants that do not confer G6PD deficiency. The authors also typed for the A- variant that confers moderate G6PD activity, however did not identify a single participant, while in contrast between 4% to 14% of participants (depending on assay applied) had phenotypic G6PD activities that would exclude them from standard radical cure for P. vivax. 

Editorial and Data Presentation Modifications?

• Parts of the manuscript need major revision for spelling, grammar, clarity and consistency.

Reviewer #2: Mostly yes except for the pieces I have called out in the comments to the authors

Reviewer #3: (No Response)

**Editorial and Data Presentation Modifications?**

Reviewer #1: Parts of the manuscript need major revision for spelling, grammar, clarity and consistency.

Reviewer #2: The manuscript needs some editorial language revision, but this is in the most part minimal.

Reviewer #3: (No Response)

**Summary and General Comments**

Reviewer #1: I am concerned about the findings from the reference method spectrophotometry. The article presents G6PD activity ranging from 0.6U/gHb to 37.7U/gHb, the latter is higher than has been reported before in humans. The distribution of activities for males should show a bimodal distribution and this is not the case, in contrast high proportion of males show an intermediate G6PD activity, which typically is associated with heterozygous females only. Interestingly the proportion of females with less than 30% G6PD activity and typically associated with homozygosity is higher than for males, again this has not been described to date and is quite unlikely. Finally, the Carestart RDT identified 14% of all samples as G6PD deficient, while only 4% were deficient by spectrophotometry. A meta-analysis on the Carestart RDT (Performance of the Access Bio/CareStart rapid diagnostic test for the detection of glucose-6-phosphate dehydrogenase deficiency: A systematic review and meta-analysis. PLoS Med) found sensitivity and specificity to be above 95%, significantly higher than the reported specificity in this article. This seems to suggest substantial issues with spectrophotometry and question many of the main findings. Can the authors discuss this in the discussion?

Reviewer #2: The authors present an interesting study assessing the G6PD prevalence in two locations in Sudan, as well as conduct an evaluation of the CareStart G6PD test. Below list some minor and some significant topics that should be addressed. 

Minor:

1. Page 6 under “Study Areas” February should be spelled correctly

2. Page 6: “The gold standard biochemical G6PD measurement was used according to a previously published methods” the authors should provide more details: for example were reagent developed internally, were they purchased, if so what kit was used?

3. Figures 1,2,and 3, the male distributions do not show clear bimodal patterns as would be expected. The authors should spend some time to explain this. How long were specimens shipped? Is there a chance there was some specimen integrity issues?

4. Top of page 7 “The two tests were performed by two independent technicians who were blinded to each one another results.” This sentence needs clarification: were two tests conducted per individual? 

5. Page 11:” AB variant was present in 1.1% (XX/557) of study subjects” XX needs to be replaced with a number

6. Page 11. Discussion: Paragraph starting “It should be emphasized that Sudan was highlighted as an area with unusually large uncertainty in the geostatistical models…” this paragraph does not seem to add anything to the discussions and should be removed.

Major

1. Table 4. Either the rows are mislabelled or the sensitivity of the CareStart is only 18% (4/22)

2. The authors should provide full performance indicators, sensitivity, specificity NPV and PPV

3. Results and Discussion regarding genotyping. The description of the genotyping is confusing, it is not clear from the genotyping what proportion of the G6PDd (n=22) were confirmed to be genotypically deficient.

4. Page 11, Discussion. Association to weight. The authors can should remove or reduce the importance of this association in their abstract, and discussion, based on the fact that this is based on 22 deficient samples and the statistical significance is weak. This is shown by the very useful set of other indicators they collected which should based on literature on humans (not mice!) have stronger associations to G6PD deficiency.

5. Discussion: genotyping “The most common G6PD genetic variant carried by subjects tested in this study is associated with a normal to moderate G6PD enzyme activity in whom drug-induced hemolysis may be self-limiting” This conclusion is a little misleading, world wide the majority of G6PD variants are normal to mild since the prevalence of G6PDd is rarely above 1%. As the authors state a limitation of this study is that the more severe G6PD variants were not investigated in this study, and so their presence cannot be excluded.

Reviewer #3: (No Response)

PLOS authors have the option to publish the peer review history of their article (what does this mean?). If published, this will include your full peer review and any attached files.

Reviewer #1: No

Reviewer #2: No

Reviewer #3: No
---

## [Decision Letter · Decision Letter 1]

9 Aug 2021

Dear Dr Abdel Hamid,

We are pleased to inform you that your manuscript 'Prevalence of glucose-6-phosphate dehydrogenase deficiency (G6PDd), CareStart qualitative rapid diagnostic test performance, and genetic variants in two malaria-endemic areas in Sudan' has been provisionally accepted for publication in PLOS Neglected Tropical Diseases.

Best regards,

Wuelton Marcelo Monteiro, Ph.D.

Deputy Editor

Marcelo Ferreira

Deputy Editor

Reviewer's Responses to Questions

**Key Review Criteria Required for Acceptance?**

**Methods**

-Are the objectives of the study clearly articulated with a clear testable hypothesis stated?

-Is the study design appropriate to address the stated objectives?

-Is the population clearly described and appropriate for the hypothesis being tested?

-Is the sample size sufficient to ensure adequate power to address the hypothesis being tested?

-Were correct statistical analysis used to support conclusions?

-Are there concerns about ethical or regulatory requirements being met?

Reviewer #2: (No Response)

Reviewer #3: (No Response)

**Results**

-Does the analysis presented match the analysis plan?

-Are the results clearly and completely presented?

-Are the figures (Tables, Images) of sufficient quality for clarity?

Reviewer #2: The figures and tables require close revision by the authors, there still seems to be some inconsistencies, for example:

The authors refer to a 4.23 u/g Hb AMM value in Table 3. but the value 4.23 is not in table 4.23

Figure 3 seems to be mis-labeled

Reviewer #3: (No Response)

**Conclusions**

-Are the conclusions supported by the data presented?

-Are the limitations of analysis clearly described?

-Do the authors discuss how these data can be helpful to advance our understanding of the topic under study?

-Is public health relevance addressed?

Reviewer #2: Some of the unexpected results observed need to be more explicitly called out

Reviewer #3: (No Response)

**Editorial and Data Presentation Modifications?**

Reviewer #2: While the article is significantly improved there remains some issues that need addressing. Unfortunately the authors did not put line numbers, and the journal did not request this.

1. Abstract: “ sites were homozygotic wild type for the G6PD G202 variant” remove the word “homozygotic” as males can only be hemizygotes and females hetero or homo-zygotes.

2. The fundamental issue is that the G6PD deficient genotype is not identified. Coupled with the fact that the distributions of G6PD activity in this population is very strange: (i) males do not show the typical bimodal distribution and (ii) there are more female deficient cases than males. This can sometimes be explained if there are issues with specimen integrity, but the authors have confirmed that the reference and the Carestart tests were performed within 30 minutes of specimen collection. These series of observations need to be explicitly called out in the discussion.

Maybe in page 9 in the paragraph starting “As expected, there were large differences in measured enzyme activity..” the authors could add ( or replace with) something along the lines of:”…The G6PD activity distributions observed in this study did not follow typical distributions observed in other studies (REF: Khim et al Malaria journal 2013, Pfeffer et al Plos medicine 2020). Specifically males did not have the expected bimodal distribution corresponding to hemizygote deficient and normal males, and more deficient females were observed than deficient males. This may result from the reference assay, but further studies are required to investigate the cause for this.

3. Figure 3 seems to be mis-labeled with G6PD normal and deficient labels being the wrong way round, otherwise as it stands the Carestart identified majority of cases as deficient and these were mostly those with high G6PD activity by the reference assay

4. Results, page 9: “…Sensitivity and negative predicative values were 91% and 99%, respectively in both sexes…” is incorrect, presumably “specificity “? Please verify and correct.

5. Results page 9: The following paragraph is confusing:

“Overall, across the two study sites, 5.5 % of the sampled population had G6PDd at an adjusted male median of 1.27 U/g Hb; 3.7% of all males and 7.1% of all females. The population adjusted male median cutoff value was 4.23 U/g Hb for both study sites. (Table 3).”

The adjusted male median value is 4.23 so the first sentence should presumably read:”… , 5.5 % of the sampled population had G6PD activity values below the 30% deficient threshold of 1.27 U/g Hb; 3.7% of all males and 7.1% of all females. The population adjusted male median value was 4.23 U/g Hb for both study sites. (Table 3).” Please also note that Table 3 does not include the 4.23 value.

Reviewer #3: (No Response)

**Summary and General Comments**

Reviewer #2: (No Response)

Reviewer #3: The authors have addressed the identified concerns and added helpful additional detail where requested.

PLOS authors have the option to publish the peer review history of their article (what does this mean?). If published, this will include your full peer review and any attached files.

Reviewer #2: No

Reviewer #3: No

---

## [Editor Report · Acceptance letter]

12 Oct 2021

Dear Dr Abdel Hamid,

We are delighted to inform you that your manuscript, "Prevalence of glucose-6-phosphate dehydrogenase deficiency (G6PDd), CareStart qualitative rapid diagnostic test performance, and genetic variants in two malaria-endemic areas in Sudan," has been formally accepted for publication in PLOS Neglected Tropical Diseases.

Best regards,

Shaden Kamhawi

co-Editor-in-Chief

Paul Brindley

co-Editor-in-Chief
